# The Association between Osteoporosis and Peripheral Artery Disease: A Population-Based Longitudinal Follow-Up Study in Taiwan

**DOI:** 10.3390/ijerph191811327

**Published:** 2022-09-09

**Authors:** De-Kai Syu, Shu-Hua Hsu, Ping-Chun Yeh, Tsung-Lin Lee, Yu-Feng Kuo, Yen-Chun Huang, Ching-Chuan Jiang, Mingchih Chen

**Affiliations:** 1Department of Orthopedics, Fu Jen Catholic University Hospital, Fu Jen Catholic University, No. 69, Guizi Rd., Taishan Dist., New Taipei City 24352, Taiwan; 2Department of Family Medicine, Fu Jen Catholic University Hospital, Fu Jen Catholic University, No. 69, Guizi Rd., Taishan Dist., New Taipei City 24352, Taiwan; 3Graduate Institute of Business Administration, College of Management, Fu Jen Catholic University, No. 510, Zhongzheng Rd., Xinzhuang Dist., New Taipei City 242062, Taiwan; 4Artificial Intelligence Development Center, Fu Jen Catholic University, No. 510, Zhongzheng Rd., Xin-Zhuang Dist., New Taipei City 242062, Taiwan

**Keywords:** peripheral arterial disease, osteoporosis, National Health Insurance Research Database, Taiwan

## Abstract

**Background:** There are several possible links that have been used to claim that osteoporosis and peripheral artery disease (PAD) are associated; however, the solid evidence is not sufficient. This study aimed to use the Taiwan National Health Insurance Research Database (NHIRD) to determine if osteoporosis is associated with peripheral artery disease (PAD). **Method:** NHIRD records from 23 million patients were collected to recruit two matched cohort groups: 64,562 patients with and 64,562 patients without osteoporosis. To compare the crude hazard ratio (HR) and the incidence rate ratio between the two cohort groups for PAD, the Cox model was used. **Result:** With an adjusted HR of 1.18 (95% CI, 1.08–1.29), the osteoporosis cohort group had a significantly greater risk for PAD than the group without osteoporosis. The cumulative incidence of PAD in the cohort group was also statistically higher than it was in the group without osteoporosis (1.71% and 1.39%; *p* ≤ 0.0001, log-rank) over the 10-year follow-up period. In addition, the osteoporotic patients with ischemic stroke, chronic obstructive pulmonary disease (COPD), and congestive heart failure (CHF) had a significantly increased risk of PAD based on subgroup analysis. **Conclusions:** There was a positive association between osteoporosis and the development of PAD, as patients with osteoporosis had an increased incidence of PAD over time.

## 1. Introduction

Peripheral artery disease (PAD) has frequently been investigated to determine its prevalence, pathophysiology, frequency and occurrence, management, and prevention in previous literature. Despite a decline in mortality rates, PAD remains highly prevalent due to the aging population, with a prevalence of approximately 10 to 20% [1]. In addition, the economic burden of atherothrombotic disease is high and is expected to increase with the aging population, demonstrating an increased survival rate. Moreover, the direct costs associated with PAD are even higher than those for coronary artery disease because of the higher rates of polyvascular disease and the higher number of annual cardiovascular (CV) events and hospitalization rates [2]. On the other hand, osteoporosis is a chronic degenerative disease that has incidence rates that increase with age. With its association with age-related fragile fractures, osteoporosis may progress without symptoms, and it is now recognized as a major threat to the elderly population. Studies are becoming increasingly focused on the links between atherothrombotic disease and osteoporosis from many aspects, including epidemiology and pathophysiology [3,4,5]. Between the mineralization of bone tissue and the calcification of the blood vessel wall, there is abundant evidence that claims their association [6]. However, while osteoporosis represents a proven link between coronary heart disease and ischemic stroke [7,8], studies investigating the relationship between osteoporosis and PAD remain controversial. There is one small cohort study that discusses the association between PAD and osteoporosis, and the results found a weak and age-dependent association between PAD and osteoporosis that was only present in women [9]. A cross-sectional study in the USA reported that the prevalence of PAD was significantly higher in women with osteopenia (4.8%) and osteoporosis (11.8%) compared to in women with a normal body mineral density (BMD) (3.3%) (*p* < 0.001). Osteopenia (OR: 1.15) and osteoporosis (OR: 1.8) were found to be independent risk factors for the presence of PAD in women [10]. On the other hand, one study conducted in the Netherlands showed that women with a low femoral neck BMD had a significantly increased risk of PAD (OR: 1.49), but this was not the case in men [11]. Although both diseases have improved our understanding of the pathophysiological factors and associations shared by advanced science, the solid evidence-based studies investigating the etiologic role of osteoporosis associated with PAD are insufficient. Therefore, the purpose of this study was to observe and analyze NHIRD records to determine if osteoporosis is associated with PAD. The hypothesis for this study was that osteoporosis is correlated with an increased incidence of PAD.

## 2. Materials and Methods

### 2.1. Data Sources

This study used the National Health Insurance Research Database (NHIRD), which is a large, computerized database that represents more than 99% of the Taiwanese population (approximately 23 million individuals) [12]. The database diagnosis codes are based on the International Classification of Diseases 9th Revision Clinical Modification (ICD-9-CM), with the 10th Revision (ICD-10-CM) being implemented in 2016. The NHIRD has often been used in previous studies identifying and using data on patients, as it includes abundant information on basic demographics, disease diagnoses, medical expenses, prescriptions, surgical operations, and medication services [13]. This study was approved by the Fu Jen Catholic University Institutional Review Board (IRB) and Ethics Committee under approval number No. C108094. Due to the NHIRD’s ethical guidelines, the personal information of the patients was anonymized before accessing the database, and thus, the IRB waived the requirements for informed consent.

### 2.2. Study Population

This research was a retrospective, case-compared, database cohort study for the evaluation of the different risks of developing PAD in an osteoporosis group and in a nonosteoporosis group. If patients were (1) older than 50 years of age and (2) had been diagnosed with osteoporosis (ICD-9-CM Code: 733.0) by a physician from 1 January 2006 to 31 December 2008, they were included in this study. If a patient had at least three different medical claims issued in an out-patient setting or at least one claim in an in-patient setting, they were defined as having osteoporosis. BMD measurement is the gold standard for osteoporosis diagnosis in the spine and hip and measures how different the mineral and calcium levels are in a particular area of the bone. A value of 2.5 or more in the standard deviation (SD) indicates osteoporosis. As shown in Figure 1, a total of 125,762 osteoporosis cases were identified after excluding 44,229 patients that were younger than 50 years old; 721 patients that had missing data; and 103,805 patients that had PAD, osteoporosis, or osteoporotic fractures, including vertebral compression fractures and hip fractures, from 2002 to 2007. A total of 2,541,021 patients without osteoporosis were included after the same exclusion criteria were applied. The nonosteoporosis cohort was verified as maintaining osteoporosis-free status throughout the entire follow-up period to minimize any allocation bias. The index date was determined from the first osteoporosis date. In order to avoid reverse causation bias, patients who were diagnosed with PAD within one year after the index date were excluded from the study. A 1:1 ratio was also used for propensity score matching along with the index year, age, and sex to reduce sampling bias. The final osteoporosis cohort contained 64,562 patients, with the comparison cohort containing the same number of patients after propensity score matching. Both cohort groups were followed up until the occurrence of PAD, death, or December 2018, whichever occurred first.

### 2.3. Outcome Measure and Confounding Variables

The incidence of PAD was the primary outcome measurement. At least three different medical statements issued in an out-patient setting or at least one claim in an in-patient setting was the definition for diagnosing PAD (ICD-9-CM codes: 440.0, 440.2, 440.3, 440.8, 440.9, 443.9, 444.0, 444.2, 444.8, 447.8, and/or 447.9). During the first year of the follow up, patients with PAD were excluded from the overall risk calculation to avoid protopathic bias. Table 1 shows that the two cohorts were balanced for index date, age, and sex after matching. Comorbidity data were collected for hypertension, hyperlipidemia, chronic obstructive pulmonary disease (COPD), diabetes mellitus (DM), liver disease, peripheral vascular disease (PVD), congestive heart failure (CHF), stroke, rheumatoid arthritis (RA), chronic kidney disease (CKD), gout, and ischemic stroke. In addition, different kinds of anti-osteoporosis drugs (including selective estrogen-receptor modulator, denosumab, bisphosphonates, and teriparatide) were added as confounding variables.

### 2.4. Statistical Analysis

The basic characteristics are presented as N (%) for the categorical variables according to the chi-square test, and the continuous variables are defined as the mean ± standard deviation (mean ± SD) according to a *t*-test. For the hazard ratios (aHRs) with 95% confidence intervals (CIs) and for the estimated incidences of the case and comparison groups, multivariate-adjusted Cox proportional hazard regression was calculated. A forest plot was applied to demonstrate the differences between the two cohorts in the subgroup analysis. Kaplan–Meier curve analyses were used to evaluate the cumulative incidences of patients with PAD, and the log-rank test was applied to determine the significance. All statistical analyses used the two-tailed *p* < 0.05 and were conducted with SAS version 9.4 (SAS Institute, Cary, NC, USA). The method used for feature selection was implemented using R software (version 3.4.3; R Foundation for Statistical Computing, Vienna, Austria).

## 3. Results

Table 1 displays the baseline characteristics and comorbidities of the study participants. There were 64,562 cases in the study and control groups. Their sex and age distributions were identical. Several comorbidities with significant differences between cohorts were found for hypertension, hyperlipidemia, chronic obstructive pulmonary diseases (COPDs), diabetes mellitus (DM), liver disease, congestive heart failure (CHF), rheumatoid arthritis (RA), chronic kidney disease (CKD), gout, and ischemic stroke. Regarding anti-osteoporosis drug usage, the osteoporosis group only had a high incidence of denosumab usage over their nonosteoporosis counterparts. On the other hand, the incidence of prescriptions of raloxifene, oral bisphosphate, zoledronic acid, and teriparatide were higher in the nonosteoporosis group than in the study group. There was no difference in ibandronate usage between both groups (*p* = 0.06).

As shown in Table 2, the osteoporosis cohort group showed a significant increase in the risk for PAD, with a crude hazard ratio (HR) and adjusted HR of 1.21 (95% confidence level (CI), 1.10–1.32) and 1.18 (95% CI, 1.08–1.29), respectively. Moreover, 1105 osteoporosis patients and 898 comparison-matched patients received diagnoses of PAD by a physician after the 10-year follow up. The incidence rate ratio was 1.19 (incidence rates of PAD per 100,000 patients per year were 172.9 in the osteoporosis cohort and 145.5 in the comparison cohort group). The crude HR was also statistically significant, showing a 21% increase in the risk of PAD in the osteoporosis group, with an HR of 1.21 (95% CI, 21.10–1.32). This significant increase in risk was sustained, even after the application of the multivariate Cox model adjusting for the aforementioned confounding factors, with an adjusted HR of 1.18 (95% CI, 1.08–1.29, *p* < 0.001).

Table 3 shows the adjusted HRs of PAD stratified by each patient characteristic in both groups using the multivariate Cox model analysis of the differential risk among the different groups. The results reveal that men and women have different levels of PAD risk, with a significant decrease in the risk for female patients (aHR = 0.658; 95% CI, 0.59–0.73; *p* < 0.001). For patients who were older in age (i.e., age 55), DM, RA, and CKD showed an increased risk of PAD compared to those who did not have these comorbidities. Regarding anti-osteoporosis drug usage, there was no significant difference in the risk of PAD. Figure 2 demonstrates the results of our multivariate Cox model analyzing the differential risk among different categorizations of PAD development stratified by the patient characteristics in the osteoporosis cohort. Osteoporosis patients with ischemic stroke had an increased risk of PAD in our study (aHR = 1.94; 95% CI, 1.40–2.67), as shown in Figure 2. In addition, the osteoporotic patients with COPD and CHF had a significantly increased risk of PAD. Our study also revealed an interesting finding that osteoporosis patients with oral bisphosphonate use had a decreased risk of PAD (aHR = 0.65; 95% CI, 0.43–0.97). On the contrary, an increased risk of PAD was noted in the study group that used ibandronate (aHR = 2.61; 95% CI, 1.08–6.31). A multivariate Cox model that excluded the follow-up events from the first year was constructed to derive the cumulative incidence function. In Figure 3, the cumulative incidence of PAD was 1.71% and 1.39% (*p* < 0.001, log-rank) for the osteoporosis and nonosteoporosis cohorts, respectively.

## 4. Discussion

Osteoporosis and atherosclerosis are two chronic degenerative diseases that share several biochemical pathways and risk factors. A previous study proved an association between osteoporosis and coronary artery disease, carotid atherosclerosis, and ischemic stroke [14,15,16]. However, studies investigating an association between PAD and osteoporosis are few and conflicting. There was a significant increase in the risk of developing PAD in patients with osteoporosis compared to their counterparts based on the multivariate Cox model (adjusted HR: 1.18). Additionally, based on the multivariate Cox model, the osteoporosis patients with ischemic stroke, COPD, and CHF had an increased risk of PAD in our study. In addition, the osteoporosis patients with oral bisphosphonate use had a decreased risk of PAD (aHR = 0.65; 95% CI, 0.43–0.97). On the contrary, an increased risk of PAD was noted in the study group that used ibandronate (aHR = 2.61; 95% CI, 1.08–6.31). Overall, a positive association between osteoporosis and subsequent PAD development was shown in this study.

The Rancho Bernardo Study [9] found a weak and age-dependent association between PAD and osteoporosis in women but not in men. Another study that had a larger sample size revealed that the prevalence of PAD was significantly higher in patients with osteopenia and osteoporosis compared to in patients with normal BMD in both genders [10]. In a large prospective cohort of 3998 Chinese men and women (65–92 years of age) in Hong Kong, the ABI correlated positively with hip BMD (correlation coefficient = 0.27; *p* < 0.001). However, after adjusting for confounding factors, the correlation was much weaker (correlation coefficient = 0.03; *p* < 0.05) [17]. This indicates that other factors may contribute to the association between peripheral vascular disease and osteoporosis. In our population-based longitudinal study, a significant increase in the risk of developing PAD was found in the osteoporosis cohort, even after adjusting for comorbidities and anti-osteoporosis drug usage. The results provide evidence of an association between the two diseases. Another current Taiwanese population-based study also used the database to compare osteoporosis and PAD [18], and they also concluded that there was a long-term risk of PAD in people with osteoporosis. There were consistent results between their study and ours, including those indicating that the PAD-free survival rate was significantly lower in the case group than in the control group. Three main differences set our study and the other study apart: First, their study used the Longitudinal Health Insurance Database 2005 (LHID2005). All of the registration and claims data for these 1,000,000 individuals were randomly sampled from the National Health Insurance Program and constitute the LHID2005. Our study used the whole population included in the NHIRD without random sampling. Second, in our study, patients who were diagnosed with PAD within one year after the index date were excluded to avoid reverse causation bias. Lastly, we included anti-osteoporosis drug usage to evaluate the drug effects.

As for the interconnections between osteoporosis and PAD, many possible pathogenesis pathways have been discussed [19]. First, when analyzing the osteoprotegerin (OPG), RANK, and the RANK-Ligand (RANKL) system, the OPG levels were significantly higher in patients with PAD than they were in patients without PAD [20]. In addition, one recent review article found that OPG levels were correlated with the presence, severity, and progression of PAD in eight articles, whereas in one article, OPG levels were not significantly elevated [21]. Second, many studies have proven that vitamin D deficiency is correlated with osteoporosis and PAD [22,23,24,25]. Third, previous studies have shown that inflammatory factors have aggravated atherogenesis and bone resorption [15]. High serum levels of C-reactive protein (CRP), interleukin (IL-6), and tumor necrosis factor-alpha (TNF-α) have all been associated with a higher risk developing PAD [26,27]. According to pathophysiologic studies, there is an association between osteoporosis and PAD. This is consistent with our study, which also provided an indication of an association. Nevertheless, a more well-designed method is needed to establish stronger evidenced-based data for a pathophysiologic association to clarify the detailed mechanisms and association.

This study has many advantages. A large sample size, a sufficient follow-up duration of up to ten years, and proper, well-balanced random matching between two cohort groups make this study valuable. However, this study has three limitations. First, due to the many physicians, specialists, and medical staff inputting data into the administrative database, possible coding errors and other coding problems may have occurred. The accuracy and coding were not verified, even though the definition of PAD and osteoporosis was based on the coding definition. However, to the best of our ability, we added two restrictions: the diagnosis (1) must have been made during two ambulatory care visits or during one or more in-patient visits and (2) needed to have been made by certain specialists. These case definitions for chronic health conditions have been widely used in the literature based on administrative data [28]. Second, NHIRD data were limited or unavailable for the proinflammatory cytokines IL-1, IL-6, and TNF-α; serial BMD value measurements; serum 25-hydroxyvitamin D; and serum parathyroid hormones; therefore, these factors could not be analyzed further. Additionally, other data, such as the patients’ BMI and lifestyle, individual dietary habits and physical activity, and data related to PAD and osteoporosis, were not recorded in the database; thus, these factors were also not analyzed. Third, although we added anti-osteoporosis drugs for analysis, different kinds of medications, such as aspirin, statin, anti-DM, and anticoagulants, were not included in the study. To eliminate the confounding factors, matching criteria may need to be added to these medications or to other interventions for propensity score calculation. A well thought-out in vivo or in vitro study should definitely be considered to analyze the pathophysiological link and causation between PAD and OP in the future since this study confirmed a positive association between the two diseases.

## 5. Conclusions

Patients with osteoporosis had a significant increased risk of developing PAD compared to patients without osteoporosis. A positive association between osteoporosis and the subsequent development of PAD could be observed. Patients with osteoporosis are recommended to receive a personalized risk screening for PAD to prevent the complications of PAD.

## Figures and Tables

**Figure 1 ijerph-19-11327-f001:**
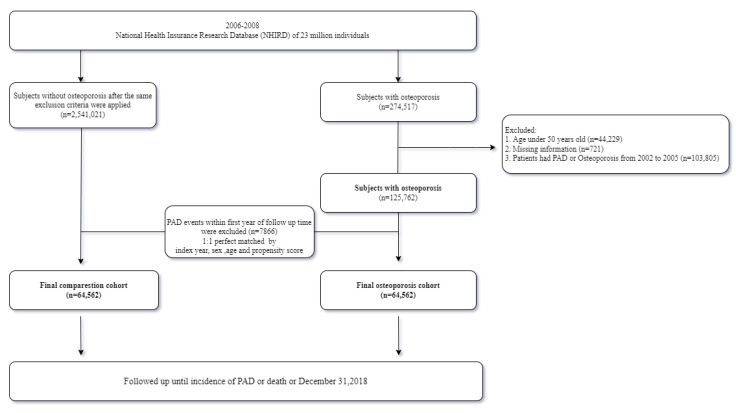
Consort diagram showing the detailed steps for assembling the two study cohorts.

**Figure 2 ijerph-19-11327-f002:**
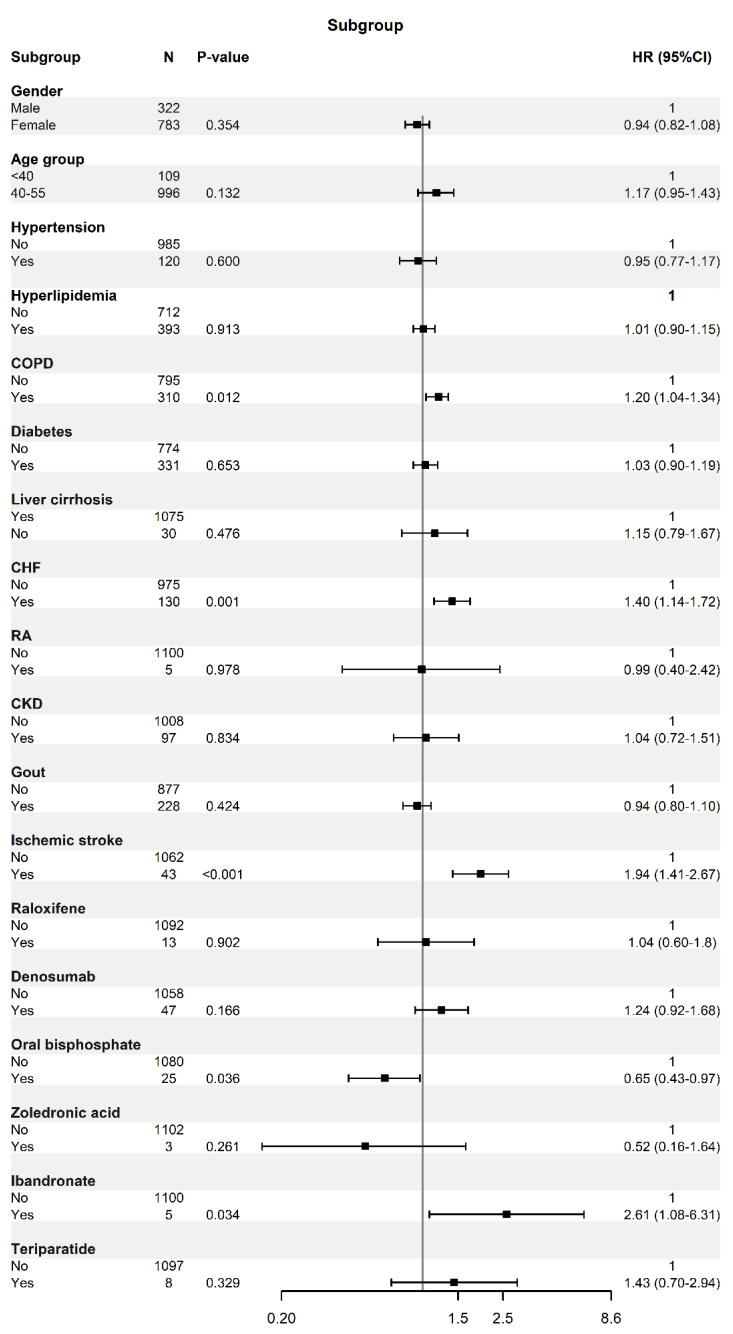
Cox model derived crude and multivariate adjusted hazard ratios (HRs) of PAD development stratified by patient characteristics in the osteoporosis cohort.

**Figure 3 ijerph-19-11327-f003:**
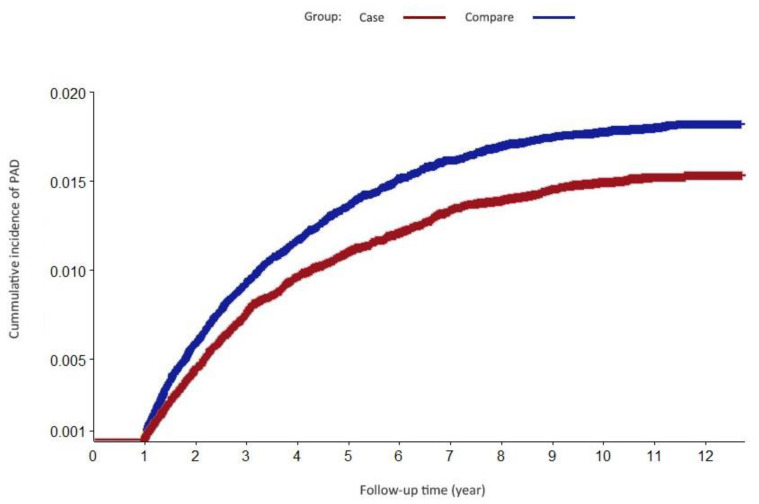
Cumulative incidence of peripheral artery diseases, which was 1.71% and 1.39%, respectively, in the osteoporosis and nonosteoporosis cohorts (*p* < 0.0001, compared to the log-rank).

**Table 1 ijerph-19-11327-t001:** Demographics and comorbidities at baseline between the osteoporosis cohort and the age-, sex-, and index date-matched comparison cohort without osteoporosis.

	Case (N = 64,562)		Comparison (N = 64,562)		*p*
	*n*	(%)	*n*	(%)	
**Year**					
2006	23,955	37.1	23,898	37.02	0.6959
2007	21,325	33.03	21,243	32.9	
2008	19,282	29.87	19,421	30.08	
**Sex**					
Female	50,454	78.15	50,555	78.3	0.4958
Male	14,108	21.85	14,007	21.7	
**Age**					
49–55	11,059	17.13	11,059	17.13	1
≥55	53,503	82.87	53,503	82.87	
Mean (SD)	66.25 (10.34)		66.25 (10.34)		1
**Baseline Comorbidity**					
Hypertension	6202	9.61	4971	7.7	<0.001
Hyperlipidemia	22,044	34.14	17,545	27.18	<0.001
COPD	18,190	28.17	11,772	18.23	<0.001
Diabetes mellitus	16,025	24.82	12,700	19.67	<0.001
Liver disease	1580	2.45	795	1.23	<0.001
CHF	6926	10.73	5130	7.95	<0.001
RA	126	0.2	87	0.13	0.0075
CKD	3969	6.15	2185	3.38	<0.001
Gout	11,227	17.39	7980	12.36	<0.001
Ischemic stroke	2809	4.35	1261	1.95	<0.001
Raloxifene	806	1.25	1024	1.59	<0.001
Denosumab	2393	3.71	1547	2.4	<0.001
Oral bisphosphate	1203	1.86	2260	3.5	<0.001
Zoledronic acid	508	0.79	623	0.96	0.0006
Ibandronate	342	0.53	391	0.61	0.0695
Teriparatide	244	0.38	338	0.52	0.0209

SD: standard deviation; COPD: chronic obstructive pulmonary disease; CHF: congestive heart failure; RA: rheumatic arthritis; CKD: chronic kidney disease.

**Table 2 ijerph-19-11327-t002:** Incidence of peripheral artery diseases and the crude and adjusted hazard ratios (HRs) derived from the Cox model for the osteoporosis cohort compared to the comparison nonosteoporosis cohort stratified by patient characteristics.

	**Case** **(N = 64,562)**			**Comparison** **(N = 64,562)**		
	N	%	Incidence Rate	N	%	Incidence Rate
PAD	1105	1.71	172.9	898	1.39	145.5
**Case vs. Comparison**						
	Crude Hazard Ratio (95% CI)		*p*-Value	Adjusted HR (95% CI)		*p*-Value
PAD	1.21 (1.10–1.32)		<0.001	1.18(1.08–1.29)		<0.001

PAD: peripheral artery disease; rate incidence per 100,000 PYs.

**Table 3 ijerph-19-11327-t003:** Overall Cox model-derived crude- and multivariate-adjusted hazard ratios (HRs) of PAD development stratified by patient characteristics in both cohorts.

	PAD	
	aHR (95% CI)	*p*-Value
Gender		<0.001
Male	1	
Female	0.658 (0.595–0.727)	
Age		<0.001
49–55	1	
≥55	1.729 (1.497–1.995)	
Hypertension		0.5842
No	1	
Yes	1.045 (0.893–1.223)	
Hyperlipidemia		0.3151
No	1	
Yes	0.949 (0.857–1.051)	
COPD		0.2621
No	1	
Yes	0.942 (0.848–1.046)	
Diabetes		<0.001
No	1	
Yes	1.291 (1.162–1.434)	
Liver cirrhosis		0.7197
No	1	
Yes	0.94 (0.669–1.32)	
CHF		0.1491
No	1	
Yes	1.12 (0.96–1.307)	
RA		0.0034
No	1	
Yes	2.664 (1.382–5.135)	
CKD		0.0029
No	1	
Yes	1.558 (1.163–2.086)	
Gout		0.1728
No	1	
Yes	1.086 (0.964–1.224)	
Ischemic stroke or SE		0.3575
No	1	
Yes	0.887 (0.687–1.145)	
Raloxifene		0.3124
No	1	
Yes	0.815 (0.548–1.212)	
Denosumab		0.2987
No	1	
Yes	1.132 (0.896–1.432)	
Oral bisphosphate		0.2789
No	1	
Yes	1.151 (0.892–1.485)	
Zoledronic acid		0.8917
No	1	
Yes	0.969 (0.612–1.533)	
Ibandronate		0.8755
No	1	
Yes	0.955 (0.538–1.697)	
Teriparatide		0.1937
No	1	
Yes	1.433 (0.833–2.465)	

## Data Availability

These data were available to us as staff of the Department of Orthopedics at Fu Jen Catholic University Hospital and of Fu Jen Catholic University using the National Hospital Research Database (NHIRD). These data are protected by the Ministry of Health and Welfare and patient privacy laws in Taiwan; no public links to these protected health information datasets are available. These data will be made available to others after receiving appropriate data privacy and human subject approval from the institution. Requests should be sent to 081438@mail.fju.edu.tw.

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
