# Peer review of "The Association between Osteoporosis and Peripheral Artery Disease: A Population-Based Longitudinal Follow-Up Study in Taiwan"

_ijerph, 2022, doi:10.3390/ijerph191811327_

Round 1

Reviewer 1 Report

The present study aimed to determine the potential association between osteoporosis and PAD, by analyzing the health data from NHIRD. Their results showed that osteoporosis cohort group had a significant increased risk of PAD. In specific, the PAD risk is gender and age dependent, as the risk is decreased in female patients, and increased in patients over 55 years old. This set of data are of interest, with merits in large sample size, systemic database, long-term follow up. However, some of the comments need to be addressed in this study.     

1. The authors stated that propensity score matching were conducted to include the subjects in each group. The authors need to provide detailed matching method in this step, such as the method used for propensity score calculation. In addition, the exclusion of 7866 cases with PAD events happened within the first year should be prior to the propensity match.  

2. Almost 43% of the cases in osteoporosis group were excluded after propensity score matching, while the sample size were almost ten times than osteoporosis group, what were the reasons for this exclusion rate?     

3. Some of the results were paradox between table 3 and figure 2, although the table 3 was targeted to both cohorts, figure 2 were only targeted to the osteoporosis cohort, what about the control cohort? The cases with PAD (incidence rate) was not uncommon as osteoporosis cohort group, how about the cox model-derived crude and multivariate adjusted hazard ratios (HRs) in control cohort?

4. Line 244, the randomization is not appropriate for this study. Propensity Score Matching is not randomized method.

5. Although the statistic analysis showed significant increased risk of PAD in patients with osteoporosis, the actual count of PAD cases is close to patients without osteoporosis, there are many confounding factors existed in two groups, including the medications, therapies. The authors need to discuss these issues in Limitation part.

Author Response

Thanks for kindly suggestion and comment.

Reviewer 2 Report

Suggestions for improvements to authors:

The article has scientific relevance and I make some suggestions to the authors:

Suggestions for improvement for authors:

The article has scientific relevance and I make some suggestions to the authors:

1. Abstract - page 1, line 28. Place the meaning of COPD and CHF before the abbreviations.

2. Summary - page 1, lines 28 to 30. In the conclusion, there is an apparent reversal of the causal factor. I suggest a change to something like "There is a positive association between osteoporosis and the development of PAD, as patients with osteoporosis have an increased incidence of PAD over time."

3. Introduction - page 2 line 62. Improve understanding of the text "There has yet to have" that was not clear.

4. Discussion - page 9, line 205. The phrase "Previous study associating PAD and osteoporosis are conflicting" was repeated from lines 195 to 196. I believe it should be removed.

5. Discussion - page 10, line 251. There is a wrong parentheses entry.

6. Conclusion - page 10, lines 263 to 268. I suggest changing the general conclusion to something like "Patients with osteoporosis have a significantly increased risk of developing PAD compared to patients without osteoporosis. The positive association between osteoporosis and subsequent development of PAD can be observed. Patients with osteoporosis are recommended to receive a personalized risk screening for PAD to avert the complications of PAD".

Author Response

(The authors gave the same response as above.)
